# Barriers and Facilitators to Physical Activity and FMS in Children Living in Deprived Areas in the UK: Qualitative Study

**DOI:** 10.3390/ijerph19031717

**Published:** 2022-02-02

**Authors:** Emma L.J. Eyre, Leanne J Adeyemi, Kathryn Cook, Mark Noon, Jason Tallis, Michael Duncan

**Affiliations:** Centre for Sport, Exercise & Life Sciences, Coventry University, Coventry CV1 2DS, UK; leannejaye@outlook.com (L.J.A.); kathryn.cook@coventry.ac.uk (K.C.); mark.noon@coventry.ac.uk (M.N.); jason.tallis@coventry.ac.uk (J.T.); michael.duncan@coventry.ac.uk (M.D.)

**Keywords:** PE, curriculum, socio-ecological model, fundamental movement skills, teacher training, determinants of health

## Abstract

Using the socio-ecological model, this qualitative study aimed to explore teachers’ perspectives on the barriers and facilitators to Fundamental Movement Skills (FMS) and physical activity engagement in children living in deprived areas in the UK. A purposive sample of 14 primary school teachers participated in semi-structured focus groups drawn from schools situated in lower SES wards and ethnically diverse areas in Central England. Thematic analysis of transcripts identified multiple and interrelated factors across all levels of the socio-ecological model for barriers to FMS and PA (i.e., intrapersonal, interpersonal, organisational, community and policy). Facilitators at three levels of influence were found (i.e., intrapersonal, interpersonal and organisational). We conclude, barriers and enablers to the PA and FMS in children from ethnically diverse backgrounds living in deprived areas are multifactorial and interrelated. At a school level, initiatives to increase PA and develop the FMS needed to be active are likely to be ineffective unless the barriers are addressed at all levels and considered more holistically with their complexity. Multi-disciplinary solutions are needed across sectors given the range of complex and interrelated factors.

## 1. Introduction

Many children in England are not engaging in enough physical activity (PA) for the associated physiological, psychological and social health benefits [1,2,3,4,5]. Given that health behaviours track from childhood into adulthood, understanding barriers and facilitators to PA in childhood is of prime importance to policymakers and public health practitioners to address this gap. PA is a multi-dimensional and complex behaviour that is influenced by physiological, psychological, social, environmental and demographic factors [6]. For children to engage in lifelong PA they require the ability to carry out Fundamental Movement Skills (FMS) (e.g., running, jumping, throwing, catching) [7]. Although FMS is posited to have a direct effect on PA during childhood, conceptual models also suggest that the effect of FMS on children’s PA may be mediated by perceived motor competence and physical fitness [8]. However, recent meta-analytical data [9] suggest that there is insufficient evidence for the suggested impact of perceived motor competence on the effect of FMS on PA. Mastery of FMS requires effective teaching of the skills as well as opportunities to practice and reinforce them in varying environments [10,11,12]. Yet, children in England have both low levels of PA [5] and FMS proficiency [13]. Of specific concern for high inactivity levels and poorer FMS attainment are children living in deprived and ethnically diverse areas in England [14,15,16,17,18]. It is suggested that between 27–65% of these children (aged 7–9 years) fail to meet physical activity guidelines [14,16] and 81.4% lack proficiency in the four required fundamental movement skills (i.e., run, jump, throw and catch) [13]. Consequently, there is a need to understand factors contributing to the low prevalence of FMS development and PA engagement in children living in deprived and ethnically diverse areas.

The barriers to PA of children living in ethnically diverse low socio-economic areas has been explored from the perspectives of parents. Despite parents perceiving the importance of their children engaging in PA [19], they identified several barriers, such as their lack of knowledge of PA, lifestyle (e.g., work commitments, other responsibilities), their local environment lacking facilities as well as their safety concerns of their neighbourhood [17,18]. Given these personal and neighbourhood barriers, parents believed schools were a key environment for their child’s PA [19,20].

The school setting is one where a child spends a large proportion of their day and where policy provides curricula for FMS. It is proposed that the school is a prime environment where programmes for promoting PA [21,22,23,24] and developing FMS are considered most effective [12]. Specifically, primary school is the first opportunity provided to every child in England to develop their FMS through the Physical Education (PE) curriculum [25]. The PE curriculum in England focuses on motor skill development from acquisition in early years (four-to-five-year-olds) to mastery and application (six-to-11-years-old) [20]. However, FMS proficiency in England since the implementation of the new curriculum is still low [13]. Teachers are responsible for the implementation of this curriculum, introducing the skills needed to engage in lifelong PA as well as providing opportunities to practice and reinforce these skills. To address the low levels observed for both physical activity and FMS, it is important to understand teachers experiences and perspectives. Having such information may provide important insights that may inform policy and practice.

To date, literature that has qualitatively explored perceptions of teachers on PA behaviours and FMS of young children living in areas of high deprivation in England is sparse. Roscoe, James & Duncan [26] explored preschool teachers’ perspectives, identifying spacious outdoor environments and outdoor equipment as facilitators of PA and developing FMS in deprived children. Preschool teachers perceived time, cost, who was responsible for PA, staff training and health and safety concerns as barriers [26]. Domville et al., [27] adopted a socio-ecological perspective to explore teachers’ (*n* = 5) perspectives of the barriers to children’s school-based PA in primary schools in a socio-economically deprived area. Domville et al., [27] identified organisation barriers (i.e., headteacher driver for PA opportunities, institutional barriers i.e., low priority of PA/PE), interpersonal barriers (i.e., poor teacher-coach relationships, individuals reducing PA opportunities), intrapersonal barriers (i.e., lack of PE training, varying teacher interest in PA and sport).

While current findings provide valuable insight into barriers and facilitators of PA opportunities, they have not included teachers’ barriers to teaching the PE national curriculum and expected outcomes. Furthermore, teachers experience of factors affecting PA and the PE curriculum in the development of FMS in deprived and ethnically diverse schools has not been considered. Given the low levels of PA and FMS in children living in deprived and ethnically diverse areas [17,18,19] and that these children engage in the majority of their PA at school [14,28], further teacher insight is needed and will offer novel insight.

Despite best efforts to improve PA and FMS in children, they may be hampered by the ability to truly understand the multiple and interrelated factors that affect both behaviours. The socio-ecological model [29,30,31,32,33,34] may provide valuable insights for addressing low levels of motor proficiency and PA engagement at critical stages to their lifelong development by understanding the multiple and interacting factors for behaviours. Ecological models postulate that behaviours have multiple levels of influence (e.g., intrapersonal, interpersonal, environmental and policy), thus gaining an understanding of the combination of psycho-social and environmental variables is best in explaining behaviours such as physical activity, PE delivery and FMS [32,33,34]. This is because the interaction with the environment is as important as what is in the environment. This approach has been used in other studies attempting to understand multiple and interrelated factors influencing PA behaviours in children [27,35]. Consequently, using the socio-ecological model, this study sought to identify the multiple and interrelated barriers and facilitators to PA and FMS perceived by teaching staff of primary schools located in a deprived and ethnically diverse area of England.

## 2. Materials and Methods

For a qualitative focus group, phenomenology research design was chosen to allow specific topic areas to be explored within the words and perspectives of teachers, their setting and complex experience, offering unique insights through their lens through moderated interaction [36]. The study adhered to the consolidated criteria for reporting qualitative research (COREQ) checklist [37] to ensure transparency in reporting of the study components.

### 2.1. Participants

Following institutional ethics approval [P45655], all participants were recruited via email using a purposive sampling approach from local primary schools situated in the most deprived electoral wards with ethnically diverse children in Coventry [38,39]. These backgrounds were targeted because the children for which they were teaching were at greater risk of inactivity and poor FMS. Focus groups included between two-to-four participants and maintained homogeneity; separated by school, year group (reception or year 5), occupation (Teacher or Teaching assistant). Following initial contact via email to the gatekeeper of the school, the interviewer met with all eligible teachers (n = 19, 4 male, 15 female) prior to study commencement to explain the research and her reasons for conducting the research (e.g., interested in improving health and wellbeing and completion of a PhD). Informed consent was provided by 14 teaching staff involved in the planning and/or delivery of the primary school curriculum or Early Years Foundation Stage Framework (EYFS) including PE in England (reception *n* = eight, all female; year 5 *n* = six, five female, one male), who took part in the study (14/19, 74% response rate). All teachers were primary qualified for their relevant post i.e., teacher or teaching assistant. In the UK primary education system, teachers are not specialists in any subjects within the national curriculum. Consequently, none of the teachers were considered PE specialists. Only the focus group participants and the interviewer were present in all sessions. Focus groups were facilitated by a Black British female, Sports Therapist with no teaching background who was in the final year of her PhD study. Teachers were aware of the facilitators background on recruitment. Prior to undertaking data collection and analysis, the researcher undertaking the main data collection and early stages of analysis undertook reflexivity to reflect on prior assumptions and experiences and how this may inform the findings. Reflexivity identified drivers for the research were gaining an understanding of low levels of activity and proficiency in England with a passion for improving health and fitness in children, specifically those from deprived and ethnic minorities. The role of parents and the home environment were believed to explain these poorer patterns. The benefit to the facilitator was a deeper understanding of the topic areas of interest as well as a contribution towards a PhD qualification. The facilitator had undergone over 50 hours of training in interviewing techniques and analysis. Throughout the interview and analysis process, the facilitator was mentored by two experienced qualitative researchers. At significant points during the process of data collection and data analysis, the researcher undertaking the data collection and early stages of the analysis, met with the wider research team with extensive qualitative experience to discuss emerging codes and categories, in interpretations of key text with similarities and differences between what the interviewer expected to find and what the data was showing. Thereby drawing combined insights in handling the data and open dialogue.

The wider research team also reflected on their backgrounds and personal anticipations. The research team consisted of White, middle class, straight, female (*n* = 2) and male (*n* = 3) academics and practitioners within physical activity. None of the authors grew up in or lived in the area being studied or are primary school teachers or teaching assistants. Two of the researchers have extensively published in the areas of physical activity and fundamental movement skills and have undertaken research projects within schools with similar demographics previously. These researchers have also undertaken qualitative work to understand barriers and facilitators with parents and children living in these areas in the past. These researchers are interested in reducing inactivity and improving the lifelong trajectories for all children. They are specifically interested in reducing the inequality gap. They have a good understanding of theoretical frameworks for fundamental movement skills and physical activity and sat independently to the derivation of coding and thematic analysis in the early stages.

### 2.2. Procedures

The method of focus groups to attain the desired information was more suitable for the current study; literature has expressed focus groups to create a more natural environment for discussion to take place compared to 1:1 interviews or questionnaires [40]. Furthermore, focus groups encouraged participants to engage in conversation, enabling them to add and respond to comments made by other participants; where they may also be prompted by one-and-others comments [40]. Focus groups were semi-structured, conducted at their workplace (i.e., school) and held at the time most convenient for participants (lunchtime or after school), lasting between 45–55 minutes. Focus group discussions were guided by two main topic areas; 1) teachers’ perceptions of FMS and PA. 2) barriers and facilitators to PA and FMS development as guided by prior qualitative work [19,20,26]. Once developed, the interview schedule was piloted with the third and last author to help the researcher identify any flaws or limitations prior to data collection [41]. Feedback from the pilot phase was used to adjust the interview schedule to reduce misunderstanding and to develop further prompts to gain further understanding [42,43]. Prior to all interviews, the interviewer followed the eight-stage interview preparation stage identified by McNamara [44] which included 1. Choosing a setting to minimise distraction; 2. Explaining the purpose of the interview; 3. Addressing confidentiality; 4. Explaining interview format; 5. Stating the length interview duration; providing researcher contact information; 7. Asking if participants had any questions before starting and 8. Asking for permission to record the interview. The trustworthiness of qualitative data is an important factor that consists of four components of credibility, transferability; dependability and conformability, which were deliberated as guided by prior research [45,46]. To improve the credibility and transferability of findings within these topic areas, all participants were required to be classroom-based staff, working directly with the children in reception or year 5, and engage in the delivery of the primary school curriculum [45,46]. Considering the dependability of the study the method was informed and guided by previous research [45,46] as well as analyst triangulation [45,47]. The main questions within each topic were open-ended and followed by several possible questions, used to probe participants, thus increasing the depth of discussion, facilitating the full expression of participant views with these topics. Probing was also conducted in response to participant’s responses to facilitate greater conformability of the data [45,46] Data was recorded using a Dictaphone (Olympus DS-2400, digital voice recorder, China) as well as making written field notes during the focus groups. Data saturation was considered when there was enough information to replicate the study [47,48], and when the ability to obtain new information is achieved and no further coding is feasible [49]. Secondary coding of the transcripts was conducted to ensure this had been reached [50]. The final sample size of 14 participants falls within the sample size guidelines suggested by Creswell [42] for phenological studies (i.e., five to 25).

### 2.3. Data Analysis

Data from the focus groups were transcribed verbatim and anonymised using ‘Fg’ [NUMBER] to indicate the focus group, ‘/R or 5’ to indicate the year group of the participant, followed by ‘P[NUMBER]’ to indicate the participant within the focus group (e.g., Fg2.1/R, P1). Transcripts were returned for comment and correction. Inductive thematic analysis was conducted firstly as described by Braun and Clarke [51] The identification of key themes thus involved a step-by-step analytical process involving data familiarisation through transcribing, reading and re-reading the data, code generation whereby short descriptive labels were assigned to the entire data set, categorisation where similar descriptive labels formed categories, searching and reviewing the themes, and defining and naming themes. This enabled a broad flexible approach to the analysis of the data collected to produce an enriched and detailed account of findings [51]. The process resulted in the formation of tabulated themes, subthemes and associated quotes. When exploring determinants of behaviour, the socio-ecological model [30] was applied in the generation of higher order themes by intrapersonal, interpersonal, organisation, community and policy level. The socio-ecological model is the most widely used ecological systems theory that considers how the person and the environment affect behaviour and how actions in one sphere influence what happens in another sphere, thus identifying the interrelated influencers to behaviour [29]. Analyst triangulation was conducted to increase quality and credibility of findings [45,46] using a second independent analyst conducted a thematic analysis which was then compared with the primary researcher; assessing potential selective perception and blind interpretive bias [45,46]. Frequent de-briefing sessions between authors facilitated the discussion, debate and re-definition of themes.

## 3. Results

The results are presented using the topic areas investigated, followed by any levels of influence in the socio-ecological model of health-related behaviour (Figure 1). The data from reception and year 5 teachers are combined unless the item was only reported within a specific teaching year group. When undertaking the focus groups, teachers commonly referred to PE when discussing FMS. The development of FMS is an implicit outcome within the PE National Curriculum in the UK. For these reasons and given the qualitative nature of the study, the subsequent results refer to PE when discussing barriers and facilitators of FMS, as this is the terminology teachers used. 

### 3.1. Teachers’ Beliefs about PA and PE

#### 3.1.1. Benefits to Health and Wellbeing

Teachers’ beliefs about PE and physical activity were positive with teachers believing that active children had improved health and well-being. Of all the PA benefits, a positive mindset was the most mentioned with children having better emotions and improved concentration and thus learning. Following a bout of PA, teachers reported:
*‘having done the mile, they are more focused when they’re back [in the classroom] they’re not as hyper’*[Fg 5.2/5, P1]
*‘I think they need that run around and space especially with so much pressure in the classroom. Their learning and concentration as well just because they will have that time to, I don’t know just run around and get on and more active’*. [Fg 2.1/5, P2]

It was further explained that active children who were skilled movers had positive early childhood experiences were more self-confident and social.
*‘the ones that have always loved PE, always done well and done stuff outside of school are already streets ahead of them. So the enthusiasm might catch up but I don’t know if the ability does’*[Fg 2.1/5, P3]
*‘they’re more social because the team playing and the games together they tend to have more friends, then play together because you know they constantly playing’*[Fg 2.2/R, P3]
*’there’re the kind of children that say to others as well, come and play’*[Fg 2.2/R, P1]

#### 3.1.2. PA and PE Important but Low Priority

Despite identifying that PE and PA were important and had many benefits, the equal importance of PE to other subjects was believed by less than half of those in the discussion. Teachers referred to its lower priority by teachers, parents, the school and policy organisation (see 3.2.3, and 3.2.4). Additionally, informal words were used to describe this movement, stating that children needed.
*‘that run around and space’*[Fg 2.1/5, P2]
*‘constantly playing’*[Fg 2.2/R, P3]

### 3.2. Factors Affecting PA and FMS in Children

Teachers identified more barriers than facilitators to PA and FMS.

#### 3.2.1. Barriers

Factors affecting PA and FMS in children were identified across all five levels of the socio-ecological model (i.e., individual, interpersonal, organisational, community, policy, Figure 1) and were interrelated barriers.

#### 3.2.2. Intrapersonal Factors

Starting with the core of the model; individual factors, children lacked the appropriate kit for PE, the ability and perception to be active or perform FMS. Teachers reported that children:
*‘don’t have PE kits’*, *‘can’t do it’ ‘they’re delayed’* and *‘don’t want to do it’* (Figure 1).

#### 3.2.3. Interpersonal Factors

Teachers and parents were key factors at the interpersonal level. The lack of time and low priority of PE and PA to other roles/responsibilities was common across both parents and teachers. For teachers, these priorities were the delivery of other core subjects which had associated performance targets and were linked to organisational and policy barriers listed later. Teachers explained;
*‘If the hall is booked people [teachers] respond with ‘oh its only PE don’t worry’’*[Fg 2.1/R, P2]
*‘PE always gets dropped off the timetable’. We haven’t finished this, we haven’t finished it, I owe you two hours of PE next week. And that’s worrying as well, some teachers, I know some teachers that have done that and it’s like they don’t mind’*. [Fg 5.1/5, P2]

Similar findings for parents were reported but their family and cultural roles influenced its value. Teachers explained:
*‘we’ve got other [parents] who don’t want to because they don’t value it ’* [Fg 2.1/5, P3], *‘some parents don’t have the time, because they are working’*[Fg 2.1/5, P3]
*‘[parents] they’re busy, the dad is always at work, never there and the mum they tend to have quite a lot of children so they’re at home on their own with these children’*[Fg 2.2/R, P3]
*‘one parent family, six or seven child family and they’ve got no-one at home to take them to these things’ [clubs]*[Fg 5.1/5, P1 & P2]

Beliefs were also important, for teachers, their lack of confidence in delivering PE and planning for PE due to lack of experience and knowledge. As a result, teachers’ identified a need for more training due to limited training during teacher training as well as CPD opportunities.
*‘I know there is a lot of teachers in the school that will purposely give PE to their PPA person, they are not confident in delivering it’*[Fg 5.1/5, P2]
*‘I wouldn’t know what would come first and then next’*[Fg 2.1/R, P2]
*‘ it’s the progression that I find difficult to find what comes next’*[Fg 2.1/R, P3]

For parents their perceptions about the safety of their local facilities, which led to parents not taking them to places for PA and FMS such as local parks. It was explained
*‘[parents] fear of even letting them outside’*[Fg 5.1/5, P2]
*‘some parents, because of the area and there’s different cultures, there’s a wide range of cultures in this area, maybe the fear of even letting them go outside’*[Fg 2.2/R, P3]
*‘they [children] will say that oh ‘parents don’t take me’, ‘there are dogs there’’*.[2.2/R, P5]

For parents, financial constraints were also a barrier to clubs and activities, with teachers stating that parents ‘*can’t pay for it, they simply don’t have the money’.* Teachers also shared how parents are interested in their children attending school clubs but if the club has a fee attached to it then they decline attendance.
*‘you do get parents being quite enthusiastic about the idea but as soon as you put a price on it, yeah, nah, no not happening’*[Fg 2.1/5, P3]
*‘they get an invoice ‘Oh no he’s not coming to lessons anymore. Because, some of them can’t pay for it they just simply don’t have the money’*[Fg 2.1/5, P3]
*‘some parents that don’t have the money because they are not working and therefore don’t have money’*[Fg 2.1/5, P3]

#### 3.2.4. Organisational Factors

At the organisation level, the school and home environment were key themes. The school curriculum organisation and delivery were reported to be crowded and influenced by performance targets related to other subjects but not PE, leaving little time for the delivery of PE, quality planning for PE as well as development of FMS and opportunities to be active within the school day. This was influenced by the policy level framework.
*‘as a class teacher you don’t have the time in your working day to then go, ‘oh they haven’t got their skills right, let’s put some quality time into doing that’*[Fg 5.1/5, P1]
*‘the work its continuous, you can’t stop…there is so many things that they have to do in the curriculum and to cover all the subjects’*[Fg 5.2/5, P2]
*‘There are not enough hours to do all the things we need to do; the marking, the planning and then allow enough time to do a quality planning of a PE lesson as well. So you know we have to try and keep that quality for that hour…you try and cut corners, look up stuff online…so that you don’t have to do the planning yourself and then tweak it to your class and do it that way’*[Fg 5.1/5, P1]
*‘Planning is on a Wednesday afternoon… we never get anything done. it’s never finished. I think because we have so much else, so much more, lots to do in the planning time, we just don’t really get to, we hardly get to theme or finish what we are supposed to be doing in the afternoon…it’s not like secondary school where you know you have teachers that, that there sole job is to maybe teach PE… We’ve got a range of subjects we have to plan for’*[Fg 2.1/5, P2]
*‘it’s just expected to be done in PPA or your own time same as anything else…maybe if we did have a bit more training, you’d have more knowledge, more experience, you’d come up with the ideas a lot quicker, you’d have a wider variety of ideas’*[Fg 2.1/5, P3]
*‘The priority is literacy and maths, closely followed by science’*[Fg 2.1/5, P3]
*‘reading, writing, maths is what you’re judged on every year’*[Fg 2.1/5, P3]
*‘they’re not going to lose literacy time to go and have their PE slot you know, if it doesn’t happen in that specific time then you know, it doesn’t happen. I’m not going to be in trouble if I haven’t got my children doing gallops and stops by the end of the year but if I haven’t got them writing sentences then I’m not going to reach my performance management tasks…so we get targets ourselves and we have to reach a certain level. So we have to have so many children that have achieved and so many children that have made good progress or excellent progress. If we don’t then that puts our jobs in a bad position. So if you’re thinking I need to get then one to write a sentence and its PE time I then well, they’re not gonna be bother about them being able to stop slowly and being able to warm up and warm down, they need to get a sentence written down I just think, well, I’ll have to’*.[Fg 2.1/R, P2]

As a result of PE’s low priority and the focus on other subjects, PE would be cancelled to catch up on other subjects or because the available facilities for PE were being used for other purposes. Teachers shared:
*‘PE always gets dropped off the timetable. We haven’t finished this, we haven’t finished it, I owe you two hours PE next week’. And that’s worrying as well, some teachers, I know some teachers that have done that and it’s like they don’t mind’* [Fg 5.1/5, P2]. *‘[The school hall may be in use] 2 weeks was morning of music …run-up to Christmas because of all the assemblies… practising the Christmas show, we can’t get in there…’*[Fg 2.1/5, P2 & P3]
*‘You get the kid’s all up there and then someone’s in there and they haven’t booked the hall or they haven’t told you’*[Fg 2.1/R, P2]
*‘If the hall is booked people respond with ‘…oh its only PE don’t worry’’*[Fg 2.1/R, P2]

Of further discussion was the employment of PE specialists with the PE funding provided by government. Teachers identified benefits, limitations and further actions for this employment. Reducing teachers teaching load, quality PE and raising the profile of PE were identified as benefits. Teachers raised concerns that specialists are trained to coach sport and not teach and thus PE was developing elitism and not becoming rounded. It was further felt that the employment was leading to teachers becoming deskilled raising concerns about the sustainability of this practice. Teachers felt a balanced approach of teacher and coach was needed where both are working together. The following examples were provided
*‘say you’ve got a 6 week half term, you start them off, maybe week 3 or 4 a real coach comes in ‘oh what have you done so far, show me’ shows us where to go next then that’s something we can carry for the rest of the half term’*[Fg 2.1/5, P3]
*‘the teacher or the TA could then be observing and watching because the PE teacher could be delivering it so just like we do for music’*[Fg 2.2/R, P3]

Teachers shared that children’s home environments provided many barriers such as lacked space to be active with small or no gardens and some living in tiny flats e.g., ‘*live in*
*flats without gardens’.* It was also reported that they stay in their house engaging with technology instead of going outside and being active for these reasons as well as associated with parents’ poor perceptions of the safety of their local facilities (e.g., interpersonal). Teachers explained:
*‘Some of them don’t let their children out in the evening …safety issues’*[Fg 2.1/5, P3]
*‘some parents, because of the area and there’s different cultures, there’s a wide range of cultures in this area, maybe the fear of even letting them go outside’*[Fg 2.2/R, P3]

At the community level, the aesthetics of the environment, norms of the community e.g., cultural, family structure, SES—classic inner city school were found to be key barriers. It was stated that:
*‘there are very few children where there isn’t something stopping them from doing as much as you would want them to, in a perfect world. Which is classic of an inner city school…some of them miss out on a lot at a young age compared to elsewhere in the country. You could drive 10 minutes down the road here and be in a completely different area. Far more affluent, far more life experience’*. [Fg 2.1/5, P3]
*‘those are the life experiences that we know are important to provide children with whereas a lot of children at this school, a lot of our children don’t get those life experiences. So that in itself is a huge barrier, you know, will ultimately become a cycle because when they’re older, they didn’t have those opportunities, will they provide them, will they know about the opportunities and they won’t provide them for their [children]. So you end up with another generation of children that aren’t physical I guess.”*. [Fg 5.1/5, P2]
*‘Physical activity is not a priority or integrated into everyday life in some cultures, a lot of the Asian and North African families it’s the go home, go to mosque, do your chores, go to bed’*[Fg 2.1/5, P3]

Finally, at the policy level curriculum policy was identified as a barrier, influencing the school environment, interpersonal factors for teachers and child level (e.g., delayed). It was reported:
*‘‘We would love to do more but the constraints of the new curriculum don’t allow it and I think in the new Profile…the new curriculum it’s like a page for KS2, that’s how much they prioritise, you know, that’s how much they hold PE up, they’ve given it a page. Whereas you’ve got like a massive document for the other subjects’*[Fg 5.1/5, P2]
*‘It’s not us, it’s getting side-lines and it’s, it is because of the curriculum that isn’t determined by us that’s determined at the government level…and it’s effecting what we do in the classroom’*[Fg 2.1/5, P3 & P2]
*‘OFSTED, Government, the data, statistics, and all that sort of stuff I think they are just happy that you’re doing PE, they don’t, you never feel like it’s a huge priority. Everything is literacy and maths now, everything else is getting pushed aside; music, art’*. [Fg 2.1/5, P3]

#### 3.2.5. Facilitators to PA and PE

Teachers reported few facilitators for PA behaviour and PE, these were identified at three levels, intrapersonal, interpersonal organisational level. For young children specifically in reception, it was felt that they had not developed a self-perception of their lack of ability that limits engagement. Teacher role modelling the activity was a key interpersonal factor, as were the provision of activities in and out of school, equipment and facilities, specialist teachers and school links with the community (organisation). Teachers shared:
*‘they go to [local sports centre] for PE and the enthusiasm that drives up, going to a proper sports hall, going to a gym, we’ve got real coaches and they all love it when they’re there’*[Fg 2.1/5, P3]

### 3.3. Teachers Training Needs

Given that teachers identified their lack of confidence and need for further training in PE, teachers were further questioned around these needs. Of the teachers interviewed, all but one had limited PE training during their teacher training.
*‘during my year PGCE I did 4 hours of PE training, so I was expected from 4 hours to deliver PE’*[Fg 5.1/5, P3]
*‘PE training at uni, when I was doing my PGCE, we had one afternoon, maybe two where we just went into a sports hall and they just showed us some athletics games…they set it all up talked to us and ‘oh have a go’. I know how to do it, I’ve been to school’*. [Fg 2.1/5, P3]

On the whole, teachers reported that the delivery method for training would be best if it allowed opportunities to observe, included having a go and some theory. There was a specific need here to be able to observe teachers/schools where children were like theirs e.g.,
*‘It would be nice to see what other schools were doing for PE… especially like [School Name] who are just around the corner... because they’ve kind of got similar children as well that would be very interesting actually to see how they do PE and how, yeah’*. [Fg 2.1/5, P2]

What teachers specifically wanted from this training was more ideas e.g.,
*‘a big overview of the whole year with something, like a breakdown of what it would look like’*.[Fg 2.2/R, P2]
*‘different ways of teaching the same thing…if children have a growth spurt they may then be imbalanced, whereas before they were pretty balanced so then they have to re-learn how to adjust to their own body, so it’s that’*.[Fg 2.2/R, P1]

However, teachers reported how needing funding and cover for their teaching limited these opportunities,
*‘there’s no funding, no-one’s gonna send us anywhere for half a day, no, they can’t afford to’*[Fg 2.1/R, P2]

## 4. Discussion

Using the socio-ecological model, the study sought to investigate the barriers and facilitators to PA engagement and FMS perceived by primary school teachers in deprived and ethnically diverse areas in England; presenting novel insights from teachers in these settings. The key findings of this study are the complexity and interrelated factors that influence PA, PE and FMS in children which are particularly challenging in deprived and ethnically diverse children. The findings further highlight the need to consider the holistic and interplay of factors in policy and practice design for sustainable change in children from these communities, a community of high risk for both development delays and lower physical activity and health outcomes [14,17,18].

### 4.1. Multiple Barriers Affecting PA and FMS

As conceptualised in the social ecological model of influences, there are many factors affecting childrens’ PA and FMS beyond the teachers’ interpersonal level (Figure 1). The interplay between these factors may provide many barriers for children from ethnically diverse backgrounds living in deprived areas. These are discussed within the sub-headings below. 

#### 4.1.1. Low Priority of PE and PA across Macro and Micro-Systems

Despite teachers believing PE was important and that being active had many benefits, the low priority of PE meant it was passed around. In the teachers’ views of PE, they reported words like ‘run around’ which may question what teachers believe PE may be, which may be affecting their perception of its priority and importance. This may be further explained by many factors reported across the interviews such as teachers lack of confidence in teaching PE, need for further training (e.g., more ideas and theory such as major and minor milestones), beliefs that PE specialists had improved the quality and profile of PE, lack of focus in their curriculum policy e.g., teachers reporting fewer pages for PE in comparison to other subjects, lack of focus for PE in their training years, less focus in school curriculum at the organisation level and performance targets for other subjects. Given the listed factors above and the focus at both policy and organisation level, it is not surprising that the teachers reported more benefits of PA and PE on academic outcomes as opposed to PE specific outcomes or physical literacy development. For example, teachers reported how a run around resulted in a greater attitude to learning, concentration and focus in subjects with performance targets. 

The low priority of PE amongst teachers [52] outsourcing PE delivery [53], lack of training, confidence and competence to teach the subject well [54] insufficient time spent during initial teacher training [55,56], lack of priority in the PE curriculum [57] have been issues reported over the last decade and more. Our study confirms that despite the improved knowledge of these barriers, the large investments in PE such as the 2012 Olympic sporting legacy and PE and sport premium [58] funding allocations, the priority of PE has not changed within the policy and school structure. Furthermore, it suggests that funding alone is not a way to address the barriers to quality PE leading to lifelong PA, due to the interplay of barriers across levels e.g., lack of priority, workload. For instance, the impact of funding can be improved by a policy change, but such changes are needed in a way that does not add to the workload of teachers. 

For example, despite additional funding for PE (Sport Premium funding) the curriculum framework that needs to be covered and the academic push and performance targets for Maths and English (i.e., reading and writing) at policy and thus school organisation level which are assessed via internal and external assessments limits time available for other subjects that are not assessed in the same way. The Office for Standards in Education, Children’s Services and Skills (Ofsted) is the department within the UK government responsible for inspecting and regulating educational services [59]. Schools in the UK are inspected and judged on their ability to meet the education requirements set by Ofsted. The school inspection handbook provided by Ofsted in 2018 [59] lists inspections of the impact of teaching literacy and maths. The handbook also states that ‘schools will not be marked down because they are not ‘tracking’ science and foundation subjects in the same ways that they may be doing in English and mathematics’ [59, pg 15). This further support of teachers’ views, indicating the lack of extensive assessment and priority placed on PE within the curriculum compared to core subjects; a view that also mirrors perceptions of head and deputy headteachers [27]. Currently, teaching staff are paid based on performance that is assessed objectively (e.g., assessment outcomes) and subjectively (e.g., through classroom observations) [60]. As assessments are seemingly more focused on English and maths, with more prescriptive guidelines and aims, the acknowledgement of performance-related pay may provide further insight into factors contributing to the priorities within the curriculum [60]. Concerns for attainment levels of disadvantaged children has resulted in additional funding at policy levels (i.e., Pupil Premium Grant) to eligible schools. The purpose of such funding is to raise attainment for disadvantaged children to be able to reach their full potential [61].

Teachers also believed that parents held a low value for PE and PA and thus PE and PA were not prioritised within these families. Prior qualitative investigations with parents of school aged (4–16 years) in low socio-economic areas and from diverse ethnic backgrounds (i.e., Asian Bangladeshi, Chinese, Yemeni), has reported the prioritisation of educational attainment over PA in these cultures [20].

#### 4.1.2. Curriculum Framework at Policy and School Level

Given the above challenges resulting in PE failing (e.g., children’s delayed movement, limited provision for PE (one-to-two lessons per week) and PE is often missed due to other demands (e.g., English and/or maths tasks, halls being double-booked)), a proposal for PE to become a core subject has been presented on behalf of the Physical Education Expert Group [62]. The grounds for this proposal include curriculum aims solely related to PE, health implications (physical and psychological) during childhood and adulthood, as well as benefits to the nation through health and economic prosperity [62]. The transition of PE from a foundation to a core subject would potentially overcome some of the barriers highlighted by teachers (e.g., low priority to core subjects, lack of quality PE planning and delivery, not meeting the recommended hours of PE) and tackle the low prevalence of PA engagement and FMS development. However, the already high demands of the current curriculum, without PE as a core subject, should be considered (e.g., high workload, assessments, planning, marking, data tracking and performance related pay). Overwhelming and unsustainable workload is the biggest reason for teachers leaving the profession and reporting a poor work to life balance, increasing stress levels and poor wellbeing [60]. Teachers already feel that the government places too much importance on data, results with a curriculum that changes too frequent, lacks opportunities to be creative, is not relevant to young people, their needs and the real world [60]. Any changes to PE would need to be considered holistically and practically within the whole education curriculum and structure and considering the challenges reported above e.g., crowded curriculum, high workload, data tracking, performance related pay. Without this holistic oversight, the changes are likely to be ineffective and just place further pressure onto a crowded, prescriptive and performance driven curriculum, negatively impacting teacher retention and wellbeing. 

#### 4.1.3. Continuing Professional Development for Teachers

In our study, the intent to up skill PE delivery was apparent with teachers identifying the need for professional training and for that to include observation, practical and theory. CPD in PE has shown to be effective but the effectiveness was hindered by short duration, limited engagement, reliance on resources as well as lack of follow up of progress [56]. Teachers’ proposals of the delivery and content for training in our study may provide solutions to the shortcomings observed by Harris, Cale and Musson [56]. Namely the duration and follow-up as teachers suggested CPD delivery every ~six or ~12-weeks as well as limited engagement which may be overcome by the incorporation of shadowing, workshops and facilitated meetings, providing opportunities to observe teachers teaching children like their children given the diverse needs and complex barriers. Given the overwhelming workload described by teachers, the barriers to effectiveness are not surprising and highlight wider issues that need to be considered when implementing the proposed solution, for example practicality of attending training due to the need for cover for CPD opportunities in an already stretched school, financial constraints, workload, need for new ideas and lack of time for planning. These again highlight that these issues need to be considered collectively to elicit effective change.

The findings about the low priority of PE, lack of provision, quality planning and lack of CPD opportunities are surprising given the large investment in PE from the government through the PE and Sport Premium allocations. Schools of 17 or more eligible pupils receive £16,000 and additional payment of £10 per pupil [61] to develop or add to the PE, PA and sports activities that their school already offers and build the capacity and capability within the school to ensure that improvements made now will benefit pupils joining the school in future years [61]. Teachers in this study identified the benefits of specialist sports coaches on reducing teachers teaching load, quality PE and raising the profile of PE, but shared concerns that it deskilled teachers and was not sustainable. This highlights that the extra funding is likely being used on areas such as using coaches to provide the provision of PE, afterschool clubs and may not be being used effectively to upskill teachers or to target those who may see the largest benefit. The use of PE and Sport Premium funding to achieve its desired aims in a crowded curriculum without consideration of these wider issues is likely to continue to be ineffective until a whole and holistic review of the curriculum framework, delivery and oversight are conducted. 

#### 4.1.4. Barriers to PA Outside of the School Environment

This study identified numerous interrelated barriers beyond the school environment affecting PA and FMS across intra- and inter-personal, organisational and community levels with certain ethnic groups facing additional barriers due to cultural and religious factors. The interaction between psycho-social and the environment were apparent, with teachers identifying parents lack of money, lack of priority/value, large family sizes and parental role within this, fears for the safety of local facilities to be active, inadequate space to be active in-home environment, community and cultural norms and what was described as the generational curse and *‘a classic inner-city school’*. Low levels of PA due to lack of perceived neighbourhood safety has been found previously linked to road safety and stranger danger [62]. In our study, these perceptions appeared to be shaped by direct experiences. Teachers reported examples of criminal activities such as drug use, vandalism, prostitution for driving perceptions that the local environments for physical activity were not safe. Furthermore, the range of cultural differences was apparent with reference to cultural prejudice between cultures which affected use of local facilities and feelings of safety. The importance of early childhood experiences to later physical activity was also identified with teachers making specific reference to a ‘generational curse’. They felt that children have negative early physical activity experiences due to their SES and cultural norms, neighbourhood barriers, resulting in a repeated cycle of behavioural patterns. Health inequalities in people from low SES backgrounds are described extensively across the literature [63,64]. Ethnicity is a key factor that determines the area for which someone resides, with the highest number of ethnic minority groups living within the most deprived areas [65,66]. Ethnic minority groups also face health inequalities which may be rooted in social, cultural and environmental factors [67]. This study provides insight from the voices in the community for the challenges faced. 

Collectively these findings support ecological models of behaviour identifying the role of psychosocial factors and environmental factors affecting behaviour as well as individual theories such as the theory of planned behaviour [68]. In the theory of planned behaviour, intention determines behaviour, which is influenced by attitudes, subjective norms and perceived behavioural control [68]. Our findings identify how external factors such as the environmental influences on PA and PE, which may be mediated through attitudes, norms and perceived behavioural control leading to subsequent intentions to be active. For example, parents did not have the intention to take children to their local parks because they perceived PA to be low priority within their culture, it not being part of their culture, they did not feel that parks were safe and felt cultural segregation (e.g., that certain parks were for specific cultures). 

These findings highlight the breadth of issues across sectors which influence PA levels and opportunities to develop FMS specifically in children in deprived and diverse neighbourhoods. A multi-disciplinary approach to address these barriers across public health, urban planning, crime prevention is needed to address these health inequalities. 

### 4.2. Limitations

The current study presents some limitations. Firstly, the study examines FMS and PA, with teachers largely focusing to discuss PE. The purpose of this paper was to focus on gaining teachers understanding of the factors that explain the lower PA and FMS proficiency levels displayed in these groups. It is important to recognise that given the focus of the PE curriculum in England to state FMS as an implicit outcome and that the focus was on teachers’ experiences, FMS was commonly discussed as PE. For these reasons, we have generated themes based on their lived experiences. However, recognising that PE is not just about preparing children for lifelong PA nor is FMS is the only prerequisite for lifelong PA. Furthermore, this study focused on the socio-ecological model in understanding behaviour, a popular theory. However, many theories exist that attempt to explain the complexity of behaviours, some which are integrated and dynamic, serving to improve understanding and identify interventions functions and techniques in population level design such as the behaviour change wheel [69]. Similarly, the PRECEDE–PROCEED model, is a participatory model for creating community health promotion [70,71]. In this study, the purpose was to gain insight into the lived experiences to understand the behaviour. In the future, if the purpose is to understand and to use that information to design community based interventions, these such frameworks may prove to be valuable. While standard processes have been followed to remove forms of conscious and unconscious bias from data collection and analysis, it is possible that unconscious bias exists which the researchers are unaware of. Despite the studies recruiting all participants regardless of their sex or ethnic background, the views presented are of mainly female, classroom-based staff, providing perceptions of those with daily first-hand experience of the topic areas explored within the study. While it would be useful to have further perceptions from male’s and other ethnic groups, these are representative of those teaching in the setting, with 75.8% of teachers in the UK being female and 85.7% white British in 2019 [72]. The findings of this study provide important insights into the challenges of low SES and ethnically diverse, however, these findings are derived from one city and warrant further investigation across cities to understand if these are unique to this city or are shared experiences. Further insights from other ethnically diverse neighbourhoods are needed to understand how findings may be representative across England. Furthermore, while the use of focus groups enabled shared experiences among the specific groups, some individuals’ unique individual beliefs may not have been voiced due to this procedure. Future insights may need to capture experiences using a range of procedures such as focus groups and interviews. Additionally, the findings of this study and earlier papers have explored independently the views of either teachers, children or parents, but none have considered collectively the perceptions of all people living in these communities as well as the views of those informing and providing policy frameworks or environmental design e.g., government, council, police as well as teachers, parents, head teachers, etc. While relatively new within this field, the collective intelligence gained from systems thinking with all stakeholders and community groups collectively and collaboratively may provide a more realistic problem solving approach to interrelated and complex factors.

## 5. Conclusions

The findings of this work provide understanding for researchers, policy leaders and stakeholders to work as collective workforces to target all levels of influence. Furthermore, the findings provide real insight into the impact of PE funding in practice, teachers PE training needs and effective ways that teachers can be upskilled in teaching PE such as facilitating observational learning with similar schools. It is apparent that for this to be successful, barriers to current practice at the multiple levels of influence need to be addressed e.g., inclusion of more training for PE in teaching courses, funding and provision at school levels to enable continuing development, working collaboratively with PE specialists and enabling time in the curriculum for good quality PE delivery. Despite schools and teachers’ best efforts to bridge the inequality gap for PA, this is likely to be ineffective unless the schooling system is considered more holistically with less priority on the prescriptive delivery of a packed curriculum and the associated reporting of these academic outcomes/attainment. A change in the schooling system which considers a holistic curriculum, a reduction in workload, further opportunities for CPD, and removal of performance related pay is likely to have a positive impact on FMS, physical activity as well as teacher and pupil wellbeing and thus may impact on teacher retention. Furthermore, the work highlights the need for collaborative working across disciplines and sectors to address the many barriers that deprived and ethnic groups face to break the cycle.

## Figures and Tables

**Figure 1 ijerph-19-01717-f001:**
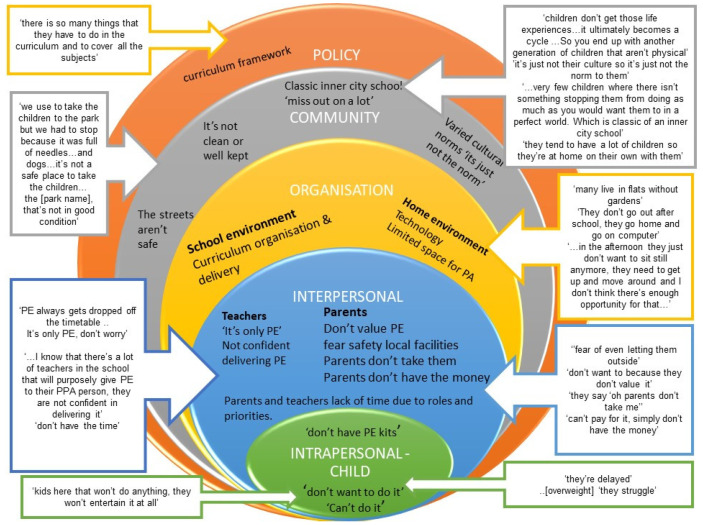
Teachers’ perspectives of barriers to FMS and physical activity in deprived and ethnically diverse neighbourhoods.

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
