# Peer review of "Barriers and Facilitators to Physical Activity and FMS in Children Living in Deprived Areas in the UK: Qualitative Study"

_ijerph, 2022, doi:10.3390/ijerph19031717_

Round 1
Reviewer 1 Report
Reviewer’s comments on:
Barriers and facilitators to physical activity and FMS in children living in deprived areas in the UK: Qualitative study
- General Comments for the Authors
The authors present interesting research on physical activity and FMS in children living in deprived areas. More specifically, if we could develop an intervention based on the results of this socio-ecological research; this could be a new step to enhance FMS and PA for these children in deprived areas.
In general, the study gives an interesting multifactorial impression of the different beliefs of the teachers. The used socio-ecological framework gives a solid overview and information for further development. However, some limitations were noted concerning the introduction and discussion. Revision is therefore required/strongly advised.
- Specific Comments
Introduction
- The introduction seems to be a combination of FMS and PA. However, some theoretical background or a model about the relationship between FMS and PA is missing. The cited article of Robinson et al. (2015) uses the model of Stodden, but also raises a lot of questions about the relationship and the role of perceived motor competence.
- It would be interesting to have some information about the PA and FMS status of the children in that area. It is assumable that they are lower, but to what extend is still a question and an important argument for this study.
- The relationship between deprived areas and PA is explained through the barriers of the parents. The rationale behind the deprived areas and FMS seems logical but an explanation is missing. Adding some background about the relation between FMS and deprived areas is therefore recommended.
- As an argument for the use of the socio-ecological model three articles are presented [28-30], but it seems that there is sparse or no information about FMS in these articles. Perhaps it’s possible to add some articles about FMS.
- The environment is an important factor. Therefore, it seems logical to ask the children or parents about their home environment/neighborhood. Either in the introduction or method section there should be some insight about the rationale to ask the teachers about this.
- Overall, there is sufficient support for the music-movement programs. There seems to be less evidence on autonomy-supporting teaching styles and music-movement programs. In other words, is there a way to discriminate the teaching style and the characteristics of the program? This can be presented in the introduction or the discussion
Methodological
- Concerning the chosen groups, it is not clear why only (female) teachers were involved.
- The selection-process is a bit misty. How many e-mails were sent and what was the respons ratio; what is the background of the teachers?
- With respect to the open nature of the questions, it would be useful if the topic list was presented as an attachment. The mentioned articles [17-19] present different methods and questions and overall, no questions were asked on FMS
Results
- The graphic presentation of the barriers gives a solid summary (some words seem missing or misplaced f.i. ‘we used xxx to the children’ and take
- The teacher’s beliefs are around PA and PE. Is there more information on FMS or is the subject actually PE (3.1)
- The general opinion is presented in the beginning of every topic, but it is unclear if this is consented by all participants or merely more than half. Providing some extra information could be presented in the results or method-section.
- There seems to be redundant information in the results (ln 260)” PE always gets drop off the timetable.” And (ln 334). Although this could be done on purpose, the difference is a bit unclear.
- If possible, give some more precise information: f.i.Teachers shared (ln 334) how many?
- There is little information about FMS (barriers and facilitators). By adding some information about FMS the results may become more balanced.
Discussion
- The information about ethnically diverse children is absent in the results. Perhaps it could be omitted in the discussion or explained in more detail?
- The suggestion to put PE in de core curriculum seems sensible, but it’s unclear what is meant by ‘ need to be considered holistically’. Does this refer to the socio-ecological levels? Perhaps an example could help.
- With respect for the concise information about the limitations, some other elements could be addressed: number of participants, other stakeholders (principal, children, parents)
- During the article, one might notice a change from FMS to PE. One might consider presenting the topic PE earlier in the introduction and perhaps even directing the whole article more towards PE.
Small remarks
- The study of Foulkes et al.(2015) addresses the low motor skills of pre-school children. Not of school children
- FMS and motor skills are used as a synonym. Although this is a detail; it would be logical to use only FMS or motor skills
- Ln 73. ethnically diverse schools are added as an extra element of deprived areas. Given the fact, there is no other information about this subject it slightly could deteriorate the reader.
- Ln 75 the children within this demographic xxxxx engage. Word is missing
- Ln 15: in deprived and ethnic children, should this be children in deprived areas? Ln 429
- 237: information missing: (see xx, and xx section).
- In 356 misspelling Live In
Author Response
Thank you for reviewing the manscript and for your feedback. Please see attachment for corrections.

Reviewer 2 Report
Thank you for the opportunity to revise this manuscript. The topic is interesting and timely.
Physical activity is vital for a child’s development and lays the foundation for a healthy and active life. Early childhood services are ideally placed to foster the development of good physical activity habits early in life. We know it is necessary a need for a greater focus on the promotion of vigorous physical activity, particularly for those children from more disadvantaged backgrounds and this manuscript help us to better explore the topic.
The introduction section should be expanded with example concerning results in the school setting with the promotion physical activity. Some examples worth including are the following:
- La Torre, G., Mannocci, A., Saulle, R., Sinopoli, A., D'Egidio, V., Sestili, C., ... & Masala, D. (2017). Improving knowledge and behaviors on diet and physical activity in children: results of a pilot randomized field trial. Ann Ig, 29(6), 584-594.
- Karppanen A-K, Ahonen S-M, Tammelin T et al. Physical activity and fitness in 8-year-old overweight and normal weight children and their parents. Int J Circumpolar Health 2012;71:1762
- D’Egidio, V., Lia, L., Sinopoli, A., Backhaus, I., Mannocci, A., Saulle, R., ... & La Torre, G. (2021). Results of the Italian project ‘GiochiAMO’to improve nutrition and PA among children. Journal of Public Health, 43(2), 405-412.
In this manuscript the socio ecological model was used but there are many models that can investigate the barriers and facilitators to improve the determinants underpinning behavior changes. Other models need to be mentioned and discussed. For example, Greeen and Kreuter introduceded in the 1970s the Precede-Proceed model, an acronym that is a good summary of the enabling, predisposing, and reinforcing factors able to change a behavior, aims to make the appropriateness of the program to the needs of the populations. Good references for this are:
- Gielen, AC, Eileen, MM.The PRECEDE-PROCEED Planning Model. In Health Behavior and Health Education, 4th ed. Jossey-Bass,1996; 51
- Parvanta, Maibach, E, Arkin, et al. Public health communication: a planning framework. Ed. Jones & Bartlett Learning 2002: 11-31
Author Response
Thank you for reviewing the manuscript and for your feedback. Please see attachment.

Reviewer 3 Report
Many thanks to the authors for this interesting study that addresses important aspects of physical activity and fundamental movement skills in children in deprived areas. The study is theoretically and empirically convincing and the results highlight the complexity of factors influencing PA and FMS as well as of the interplay between them. In some respects, however, the argumentation is somewhat vague and should be sharpened. Further, some methodological aspects remain unclear. Hence, some revisions would enhance the quality of this paper. These are outlined below.
- Even though the complex relationship between PE and PA is addressed in the paper, elementary aspects remain unclear or unspoken. First, as a school subject with educational aspirations, physical education is much more than "just" PA. Therefore, the implicit equation of PE and PA, which pervades the interviews as well as parts of the argumentation, is not tenable and at least critical in the discussion. Second, PE is also more than providing kids with skills for lifelong PA. PE should rather introduce learners to a self-determined and responsible approach to physical activity and their own health in the sense of physical literacy. Children are not trivial machines that need to be equipped with skills and physically activated for life. Rather, they are independent persons who are to be introduced to a self-reliant way of life.
- The decision to use focus groups is explained in principle in the paper. However, this approach also has its limitations. In qualitative research, focus groups are often used to explore shared knowledge and to find out what is generally understood in a particular group of people as something that can be said and what cannot. Therefore, in this approach, things that are thought of by individuals but understood as something that cannot be said tend to remain in the background. This should be reflected as a limitation of the study.
- The researchers should share some information about their thoughts, beliefs and expectations regarding PA, FMS and children living in deprived areas – as this informs the research process in a qualitative study – and they should critically reflect on their involvement in the process. Although the goal was to allow for the highest level of quality and credibility in the findings, the research findings remain the result of a rather subjective process. In this context, moreover, one might also ask what the researchers did when they assessed “potential selective perception and blind interpretative bias” in the analysis?
- Few typos should be corrected (like brackets and numbers missing).
Author Response
Thank you for reviewing the manuscript and your constructive feedback. Please see attachment.

Reviewer 4 Report
This study lay the foundations of a strong for barriers and enablers to the PA and FMS. The study is innovative and addresses important information on the area of deprived and ethnic children. Even though, the manuscripts present some flaws that must be considered. I recommend its publication after minor changes.
- The general objective and specific objectives should appear at the end of the introduction. The objective should be clearly written, referring to the population, the intervention, the comparison and the results (PICO strategy).
- The sample is very small for this type of study that should pay careful attention to its inference results, and should be limited in the article.
Author Response
Thank you for reviewing the manuscript and your feedback. Please see attachment.

Round 2
Reviewer 1 Report
all comments are taken adequately care for. A constructive way to respond to a reviewer. Thanks